# *Lonicera japonica* Fermented by *Lactobacillus plantarum* Improve Multiple Patterns Driven Osteoporosis

**DOI:** 10.3390/foods13172649

**Published:** 2024-08-23

**Authors:** Zimin Chen, Weiye Xu, Jianming Luo, Liu Liu, Xichun Peng

**Affiliations:** Department of Food Science and Engineering, Jinan University, Guangzhou 510632, China; 13107624457@163.com (Z.C.); xweiye@126.com (W.X.); luojm@jnu.edu.cn (J.L.); cdliuliu@126.com (L.L.)

**Keywords:** fermented *Lonicera japonica*, bone metabolism, rat experiment, serum metabolites, intestinal flora

## Abstract

Osteoporosis (OP) represents a global health challenge. Certain functional food has the potential to mitigate OP. Honeysuckle (*Lonicera japonica*) solution has medicinal effects, such as anti-inflammatory and immune enhancement, and can be used in functional foods such as health drinks and functional snacks. The composition of honeysuckle changed significantly after fermentation, and 376 metabolites were enriched. In this study, we used dexamethasone to induce OP in the rat model. Research has confirmed the ability of FS (fermented *Lonicera japonica* solution) to enhance bone mineral density (BMD), repair bone microarchitectural damage, and increase blood calcium levels. Markers such as tartrate-resistant acid phosphatase-5b (TRACP-5b) and pro-inflammatory cytokines (TNF-α and IL-6) were notably decreased, whereas osteocalcin (OCN) levels increased after FS treatment. FS intervention in OP rats restored the abundance of 6 bacterial genera and the contents of 17 serum metabolites. The results of the Spearman correlation analysis showed that FS may alleviate OP by restoring the abundance of 6 bacterial genera and the contents of 17 serum metabolites, reducing osteoclast differentiation, promoting osteoblast differentiation, and reducing the inflammatory response. This study revealed that *Lactobacillus plantarum*-fermented honeysuckle alleviated OP through intestinal bacteria and serum metabolites and provided a theoretical basis for the development of related functional foods.

## 1. Introduction

Osteoporosis (OP) is the most common metabolic bone disease, characterized by reduced bone mass and deteriorating bone microstructure due to an imbalance between bone formation and resorption [1]. The International Osteoporosis Foundation (IOF) reported that OP affects over 200 million people globally and causes a fracture every three seconds [2,3]. Since the 1990s, hip fracture incidence has significantly increased, with predictions suggesting a rise of 240% in women and 310% in men by 2050 [4]. Current OP treatments face challenges with efficacy and safety. Selective estrogen receptor modulators show potential but are limited by side effects [5]. Anti-resorptive agents like bisphosphonates reduce fracture risk by 50% but raise concerns about long-term use and patient adherence [6]. Thus, exploring functional foods to alleviate OP symptoms and enhance bone density is promising.

Honeysuckle (*Lonicera japonica*) is renowned for its medical and nutritional properties, it is rich in polysaccharides, polyphenols, flavonoids, and other bioactive compounds. Notable phenolic acids like chlorogenic, caffeic, and ferulic acids are highlighted for their OP-treating potential [7,8,9]. Honeysuckle flavonoids, such as luteolin, quercetin, kaempferol, and rutin, may also offer therapeutic benefits for OP [10,11,12,13]. Plant polysaccharides, such as tea polysaccharides and astragalus polysaccharides, are increasingly recognized in improving OP [14,15]. Tea polysaccharides inhibit osteoclast buildup and enhance bone structure in osteoporotic rats [14]. Astragalus polysaccharide increases bone density and reduces TNF-α levels, showing anti-OP effects [15]. Honeysuckle’s potential to restore OP may stem from the synergistic action of multiple components.

The gut microbiota (GM), a mix of various microorganisms in the human gut, is vital in health and disease [16]. It supports bone health via the gut–bone axis, impacting the metabolic, immune, and hormonal systems [17]. Gut microbiota-produced metabolites, such as short-chain fatty acids, indole derivatives, polyamines, and adenosine triphosphate, are crucial in bone metabolism [18,19]. Additionally, GM influences the immune system, with immune cells like Treg and Th17 cells playing a key role in bone health [20,21]. The gut microbiota also engages in endocrine interactions, producing hormonal substances like 5-HT and estrogen, which are essential for bone metabolism regulation [22,23].

*Lactobacillus plantarum* (LP), a key strain in the *Lactobacillus genus*, is commonly found in various fermented foods. LP attenuates glucocorticoid-induced osteoporosis by altering the composition of the rat gut microbiota and serum metabolic profile [24]. Fermentation, a traditional practice in Chinese Medicine, enhances treatment efficiency, reduces toxicity, and generates beneficial metabolites (organic acids and peptides, etc.) [25]. As a widely used strain in modern traditional Chinese medicine, LP modulates the gut microbiome and offers various therapeutic benefits (reduces toxicity and modulates immune function, etc.) [26].

There are few reports on honeysuckle in OP treatment, and none on LP-fermented honeysuckle. This study aims to explore how LP-fermented honeysuckle can more effectively relieve OP, by examining shifts in colonic flora and serum metabolites.

## 2. Materials and Methods

### 2.1. Making Honeysuckle Extract

The honeysuckle (*Lonicera japonica*) used in this experiment was sourced from Haozhou, Anhui, China (March 2023). First, honeysuckle buds were crushed and passed through a 60-mesh sieve to obtain the honeysuckle powder. Then, it was twice-boiled in water (25:1 material-to-liquid ratio) for an hour. The liquid was filtered, condensed, and freeze-dried to produce the final sample.

### 2.2. Quantification of Honeysuckle’s Crude Polysaccharides

Initiating with the dissolution of the samples in water, 4-fold (*v/w*) absolute ethanol was subsequently added. After a preservation period of 12 h at 4 °C, the product underwent centrifugation (4000× *g*, 10 min), retaining the precipitated polysaccharide that was washed thrice with ethanol. This resulted in a precipitate that was collected via vacuum freeze-drying. Following additional dissolution, concentration, and freeze-drying, the honeysuckle’s crude polysaccharides were obtained. The phenol–sulfuric acid method used to measure this entirety indicated that the polysaccharides constituted 9.76% of the honeysuckle’s total composition (standard curve: y = 3.315x + 0.04667, R^2^ = 0.9981) [27].

### 2.3. Preparation of Fermentation and Mixed Solution

The fermentation of the medium formulation in our laboratory was as follows: skimmed milk powder 1 g/L, glucose 6 g/L, aqueous extract 256 g/L, and bacterial strain addition 8% (*v/v*). The above solution was fermented at 37 °C for 20 h and stored at 4 °C. In order to analyze the differences between honeysuckle before and after fermentation. An unfermented solution, namely mixed solution (MS), was used as a control.

### 2.4. Animal Experiments

Thirty specific pathogen-free (SPF) female Sprague Dawley (SD) rats, aged 7–8 weeks, were sourced from the Guangzhou Weitonglihua Company (Guangzhou, China). They were housed in an SPF-grade room, temperature-controlled at 23 ± 2 °C and humidity-controlled at 55% ± 5%, in a 12-h light–dark cycle (light on from 8:00 to 20:00). Standard chow AIN-93M and sterile distilled water were provided during the experiment, and the rats had free access to food and water. After a week of acclimatization, they were divided into five groups (each containing 6 rats) and administered different treatments: the Con group received normal saline injections into the leg muscles twice a week and were orally gavaged with sterile water once daily. The other groups were administered injections of dexamethasone into the leg muscles at a dosage of 0.1 mg/100 g. The Hse group were orally gavaged with the honeysuckle extract solution at a dosage of 1 mL/100 g daily. Similarly, the Fer group were orally gavaged with the fermentation solution at the same dosage of 1 mL/100 g daily. The Mix group were orally gavaged with a mixed solution at a dosage of 1 mL/100 g daily. Lastly, the Mod group were administered injections of sterile water at a dosage of 1 mL/100 g daily. Nine weeks later, all the rats were fasted overnight before being euthanized and anesthetized with sodium pentobarbital (40 mg/kg). Blood was collected from the abdominal aorta, and serum was obtained after centrifugation. The serum was analyzed for various parameters. Tibias and femurs were collected for morphological studies and bone mineral density (BMD) testing, respectively. The contents of the colon were collected for 16S rRNA sequencing and analysis.

The Ethics Committee of Jinan University approved all the animal studies (Approval No. IACUC-20230522-20), and all the guidelines were rigorously followed.

### 2.5. Bone Mineral Density (BMD) Assessment

The BMDs of the femurs were determined using dual-energy X-ray absorptiometry (Lunar iDXA scanner (GE Healthcare, Chicago, IL, USA)) under the following parameters: 0.0188 A, 100 kV, and 10.0 μGy.

### 2.6. Analysis of the Tibia Paraffin Section, H&E Staining, and Bone Microarchitecture

Post fixation in 4% paraformaldehyde for 3–5 days, the tibia underwent decalcification, dehydration, extraction, and subsequent paraffin permeation. Paraffin-embedded samples were sectioned at a thickness of 5 μm, followed by hematoxylin and eosin (H&E) staining. To analyze the sections, a Pannoramic MIDI digital scanner (3DHISTECH Ltd., Budapest, Hungary) was employed. The slides, once mounted with neutral balsam and covered with coverslips, were scanned. The Image Pro Plus 6.0 software facilitated the calculation of the tissue area (T.Ar), the trabecular bone area (Tb.Ar), and the trabecular bone perimeter (Tb.Pm). The computed parameters included the bone volume/total volume (BV/TV) ratio, the trabecular number (Tb.N), and the trabecular separation (Tb.Sp), following the equations: BV/TV (%) = Tb.Ar/T.Ar × 100%; Tb.N(mm^−1^) = (1.199/2) × (Tb.Pm/T.Ar); Tb.Sp(μm) = (2000/1.199) × (T.Ar − Tb.Ar)/Tb.Pm [28].

### 2.7. Assessment of Blood Calcium, TRACP-5b, OCN, TNF-α, and IL-6

Using ELISA kits supplied by the Nanjing Jiancheng Bioengineering Institute Co., Ltd., Nanjing, China; the levels of blood calcium, TRACP-5b, OCN, TNF-α, and IL-6 were quantified in the rat serum acquired post centrifugation, adhering to the manufacturer’s instructions.

### 2.8. Sequencing of 16S rRNA in Colon Content Microbiota

The colon contents were collected fresh and kept at −80 °C after the rats were sacrificed. The microbial DNA were isolated and amplified using 338F (5′-ACTCCTACGGGAGGCAGCAG-3′) and 806R (5′-GGACTACHVGGGTWTCTAAT-3′) of the bacterial V3-V4 region of the 16S rRNA gene (PCR). The 20 μL reaction system consisted of 4 μL of 5× FastPfu Buffer, 2 μL of 2.5 dNTPs (2.5 mM), 0.8 μL of Forward Primer (5 μM), 0.8 μL of Reverse Primer (5 μM), 0.4 μL of FastPfu Polymerase (0.4 μL), 0.2 μL of BSA, 10 ng of template DNA, and the proper amount of ddH2O to the final volume. The amplifications were carried out as follows: initial denaturation at 95 °C for 3 min; 27 cycles of denaturing at 95 °C for 30 s, annealing at 55 °C for 30 s, and extension at 72 °C for 45 s; extension at 72 °C for 10 min, and ending at 4 °C. The PCR products were purified after being tested on a 2% agarose gel electrophoresis. The 16S rRNA amplicons were pooled in equimolar and paired-end sequences using an Illumina MiSeq PE300 platform/NovaSeq PE250 platform (Illumina, San Diego, CA, USA) according to the standard protocols by Majorbio Bio-Pharm Technology Co., Ltd. (Shanghai, China). Usearch (version 7.1, http://drive5.com/uparse/ 23 December 2023) was used to perform the clustering analysis on sequences with a 97% similarity level, and the sequences were designated as operational taxonomic units (OTUs). The microbiota diversity, sample comparison, and gut microbiota profile were examined after the taxonomic analysis of representative OTU sequences. The Mothur software (version v.1.30, http://www.mothur.org/ 15 January 2024) was used to assess the alpha diversity index for each sample, and then a student’s *t*-test was used to evaluate the statistic differences among groups.

### 2.9. Metabolite Extraction and LC-MS/MS Analysis

We performed a metabolite analysis in FS, MS, and rat serum. A 50 mg sample was accurately weighed into a 2 mL centrifuge tube, 400 μL of extraction solution [methanol:water (v:v) = 4:1] containing 0.02 mg/mL of the internal standard (L-2-chlorophenylalanine) were added, and then ground for 6 min (−10 °C, 50 Hz) in a frozen tissue mill, followed by 30 min of low-temperature ultrasound extraction (5 °C, 40 KHz). After the extraction, the samples were left at −20 °C for 30 min and then centrifuged for 15 min (13,000× *g*, 4 °C), and the supernatant was then taken for a liquid–liquid mass spectrometry analysis. In addition, 20 μL of supernatant were removed from each sample and mixed as the quality control sample. The chromatographic conditions were as follows: column ACQUITY UPLC HSS T3 (100 mm × 2.1 mm i.d., 1.8 μm; Waters, Milford, MA, USA), 95% water +5% acetonitrile (containing 0.1% formic acid) as the mobile phase A, 47.5% acetonitrile +47.5% isopropanol +5% water (containing 0.1 % formic acid) as the mobile phase B, 0.40 mL/min of the flow rate, 10 μL of the injection volume, and 40 °C of the column temperature. The gradient elution parameters were set as follows: 0–0.5 min, 0% B; 0.5–2.5 min, 0~25% B; 2.5–9 min, 25~100% B; 9–13 min, 100% B; 13–13.1 min, 100~0% B; 13.1–16 min, 0% B. The samples were ionized using electrospray ionization, and the mass spectrum signals were acquired in the positive and negative ion scanning modes, respectively. The scanning range was 50–1000 *m/z*, the ionization voltage was (positive) 5000 V, the ionization voltage was (negative) −4000 V, the declustering voltage was 80 V, the spray gas was 50 psi, the auxiliary heating gas was 50 psi, the gas curtain gas was 30 psi, the heating temperature of the ion source was 500 °C, the cycle collision energy was 20–60 V, and the cycle time was 510 ms. Quality control (QC) samples were prepared by mixing the equal volume extracts of all the samples. Each QC sample, with the same volume of the analytical samples, was processed and assayed in the same way as the analytical samples. During the instrumental analysis, one QC sample was inserted into every 5–15 analytical samples to examine the stability of the entire assay process.

After the samples were uploaded, the raw LC-MS data were imported into the metabolomics processing software Progenesis QI 2.0 (Waters Corporation, Milford, MA, USA) for peak detection and alignment. The preprocessing results generated a data matrix that consisted of the retention time (RT), mass-to-charge ratio (*m/z*) values, and peak intensity. Metabolic features detected at least 80% in any set of samples were retained. After filtering, the minimum metabolite values were imputed for specific samples in which the metabolite levels fell below the lower limit of quantitation, and each metabolic feature was normalized using the sum. The internal standard was used for the data QC (reproducibility). The metabolic features for which the relative standard deviation (RSD) of QC was >30% were discarded. After the standardized procedures and imputation, the log conversion data were statistically analyzed to determine significant differences in the metabolite levels between comparable groups. The mass spectra of these metabolic features were identified using the accurate mass and MS/MS fragments spectra; the isotope ratio difference was obtained by searching in reliable biochemical databases, such as the Human metabolome database (HMDB) (http://www.hmdb.ca/ 15 January 2024) and the Metlin database (https://metlin.scripps.edu/ 15 January 2024).

The pre-processed data were uploaded on the Majorbio cloud platform (https://cloud.majorbio.com 15 January 2024) for data analysis. The R package (Version 1.6.2) was used for the principal component analysis (PCA) and the orthogonal least squares discriminant analysis (OPLS-DA), and 7 round-robin interaction validation was used to assess the model’s stability. In addition, a student’s *t*-test and multiplicative analysis of variance were performed. The selection of the significantly different metabolites was determined according to the variable weight value (VIP) obtained from the OPLS-DA model and the student’s *t*-test *p*-value; metabolites with VIP > 1 and *p* < 0.05 were considered as significantly different metabolites.

### 2.10. Statistics

All data are presented as mean ± standard deviation (SD) values. The Spearman correlation analysis was performed using the Majorbio Cloud Platform online tool. Statistical differences were analyzed using a one-way analysis of variance (ANOVA) or the independent *t*-test using the GraphPad Prism 9 software (La Jolla, CA, USA). The data from different groups in the independent experiments were considered statistically significant when *p* < 0.05.

## 3. Results

### 3.1. Effect of FS on Bone Structure

#### 3.1.1. FS Significantly Restored BMD Levels

BMD serves as a crucial metric for the diagnosis of OP [29]. Figure 1 illustrates that the BMD for the Mod group (0.2275 g/cm^2^) was substantially inferior to the BMD of the Con group (0.2873 g/cm^2^). These observations testify to the significant bone loss in the Mod group. The observed BMD within the Fer group (0.2673 g/cm^2^) escalated to a level similar to the Con group (Figure 1). Additionally, it was significantly superior compared with the Mod group, the Hse group (0.2508 g/cm^2^), and the Mix group (0.2567 g/cm^2^) (Figure 1). The findings highlight that FS has brought a remarkable improvement to the BMD levels in rats with OP.

#### 3.1.2. FS Significantly Promoted the Recovery of Bone Microstructure

Impairment to the bone microstructure is a dominant characteristic of OP [30]. Figure 2 exhibits that, in comparison with the Con group (Figure 2A), the Mod group (Figure 2B) demonstrated fewer bone trabeculae with a sparse disposition. These trabeculae appear small and thin, forming a localized network with a large number of gaps. These observations testify to the significant bone loss in the Mod group.

Significant restorations were observed in the number, size, thickness, and spatial distribution of the bone trabeculae in the Fer group (Figure 2E). This recovery seemed more pronounced than that of the Hse group (Figure 2C) and the Mix group (Figure 2D). Appendix A indicates that in comparison with the Con group, the Mod group recorded substantial reductions in Tb.Ar and BV/TV, while T.Ar, Tb.Pm, and Tb.N exhibited a decreasing trend, and Tb.Sp increased significantly (Appendix A). However, when compared with the Mod group, Tb.Ar and BV/TV in the Fer group increased notably, while T.Ar, Tb.Pm, and Tb.N demonstrated a trend towards recovery, and Tb.Sp significantly decreased. Furthermore, when compared with both the Hse and Mix groups, the Fer group showcased superior recuperation (Appendix A). Accordingly, the findings reveal that FS significantly bolstered the recuperation of the bone microstructure.

### 3.2. Effect of FS on Some Related Factors in Serum

#### 3.2.1. FS More Significantly Improved Calcium Absorption in OP

Reduced blood calcium concentrations signify OP [31]. As depicted in Figure 3, the Mod group presented a significantly lower blood calcium concentration (1.691 mmol/L) compared with the Con group (2.004 mmol/L). Notably, the Fer group witnessed a significant elevation in the blood calcium concentration (1.922 mmol/L) (Figure 3). Nevertheless, the Hse group (1.710 mmol/L) and the Mix group (1.793 mmol/L) did not display any marked divergence from the Mod group (Figure 3). These observations confirm that FS significantly restored the serum calcium level in OP rats.

#### 3.2.2. FS Significantly Reduced the Osteoclast Differentiation Biomarker TRACP-5b and Increased the Osteoblast Differentiation Marker OCN

TRACP-5b and OCN serve as crucial differentiation markers for osteoclasts and osteoblasts [32]. Referring to Figure 4A, the Mod group recorded higher levels of TRACP-5b (3.907 ng/mL) compared with the Con group (2.562 ng/mL). In addition, the OCN level in the Mod group (32.00 ng/mL) fell notably compared with the Con group (37.87 ng/mL), as illustrated in Figure 4B. This indicates a noteworthy enhancement in osteoclast differentiation and a notable inhibition of osteoblast differentiation within the Mod group. The Hse and Mix groups’ TRACP-5b (3.076 ng/mL, 2.421 ng/mL) and OCN (37.19 ng/mL, 37.57 ng/mL) levels were revived, nearly matching the Con group’s levels (Figure 4). Moreover, the Fer group demonstrated superior recuperation of TRACP-5b (1.739 ng/mL) and OCN (42.74 ng/mL) compared with the Hse group and the Mix group (Figure 4). The results suggest that FS substantially curbed osteoclast differentiation and fostered osteoblast differentiation.

#### 3.2.3. FS More Significantly Reduced the Levels of the Pro-Inflammatory Cytokines TNF-α and IL-6

Proinflammatory cytokines, including TNF-α and IL-6, pose a high risk for OP [33]. As evidenced by Figure 5, the Mod group’s TNF-α concentration (244.8 ng/L) marked a notable rise relative to the Con group (178.2 ng/L), whereas it noticeably declined in the Fer group (200.6 ng/L). The Fer group’s TNF-α level demonstrated a more impactful recovery when compared with that of the Hse and Mix groups (219.3 ng/L, 225.4 ng/L) (Figure 5A). Likewise, similar to TNF-α, the IL-6 level in the Mod group (87.78 ng/L) increased significantly compared with the Con group (31.89 ng/L), whereas the Fer group indicated a notable decrease (55.98 ng/L). The Fer group demonstrated a more remarkable recovery of the IL-6 level versus the Hse and Mix groups (81.09 ng/L, 69.97 ng/L) (Figure 5B). The results show that FS significantly reduced the secretion of the pro-inflammatory cytokines TNF-α and IL-6.

### 3.3. Effect of FS on the GM of OP Rats

#### 3.3.1. FS Intervention Significantly Changed the GM of OP Rats

The results of the alpha diversity indicate a diminished diversity in colonic microbiota within the OP (Figure 6). This trend was mitigated by the implementation of treatments. Such treatments contributed to a higher Chao value in the three treatment groups than the Mod group (Figure 6A), whereas the Simpson index was significantly lower within the three treatment groups, even lower than that in the Con group (Figure 6B). Consequently, FS has been observed to effectively restore the diversity of GM, which was destroyed by OP. As suggested by the partial least squares discriminant analysis (PLS-DA), there was a clear demarcation among the groups, with minimal intersections existing between the Mix and Fer groups (Figure 6C). This implies distinct variations in the GM composition across these groups.

#### 3.3.2. FS Improves OP by Restoring the Abundance of Possible Key GMs

At the phylum level, the *Firmicutes phylum* was dominant (Figure 7A). A genus level analysis identified six bacteria that may be key to OP recovery in FS (Figure 7B). Compared with the Con group, the Mod group showed a significant decrease in the relative abundance of six bacterial genera, including *Lactobacillus*, *unclassified_f__Oscillospiraceae*, *Ruminococcus_torques_group*, *norank_f__Peptococcaceae*, *Candidatus stoquefichus*, and *Family_XIII_AD3011_group*, and a significant increase in the prevotellaceae_Ga6A1_group (Appendix A). These changes were restored in the Fer group (Appendix A).

Consequently, we inferred that the prevotellaceae_Ga6A1_group could be involved in the pathogenesis of OP. These six bacterial genera (*Lactobacillus*, *unclassified_f__Oscillospiraceae, Ruminococcus_torques_group, norank_f__Peptococcaceae, Candidatus stoquefichus*, and *Family_XIII_AD3011_group*) may be the key bacteria in reducing OP by FS.

### 3.4. Influence of FS on Rat Metabolite Levels

#### 3.4.1. Comparison of Metabolites between FS and MS

The PLS-DA graph demonstrated a significant alteration in the metabolic components of honeysuckle pre-fermentation and post-fermentation (Figure 8A). As illustrated in the differential metabolite volcano plot, in comparison with MS, FS led to the upregulation of 376 metabolic constituents and the downregulation of 419 (Figure 8B).

#### 3.4.2. Effects of FS on Serum Metabolites in OP Rats

A spectrum of 337 differential metabolites was observed in the serum metabolome data between the Con and Mod groups, with the Mod group noted for downregulating 192 of these serum metabolites while upregulating 145 (Figure 9A). Among the 232 differential serum metabolites identified between the Mod and Fer groups, the Fer group upregulated 106 serum metabolites and downregulated 126 (Figure 9B). In contrast, 380 differential serum metabolites were found in the Mod and Mix groups, with the Mix group increasing 178 serum metabolites and decreasing 202 (Figure 9C). Furthermore, in the Mod and Hse groups, 118 serum metabolites were found to be upregulated and 133 downregulated among the 251 differential serum metabolites detected in the Mix group (Figure 9D).

It is noteworthy that the relative abundance of 17 metabolites decreased and the relative abundance of 23 metabolites increased in the serum of the Mod group (Appendix A). The levels of these metabolites were significantly restored after treatment in the Fer group. The recovery of these 40 serum metabolites may be associated with the repair and pathogenesis of OP.

### 3.5. Correlation Analysis between Bone Markers and Possible Key Bacterial Genera and Possible Key Metabolites

A correlation analysis between the possible key bacterial genera and bone markers. The 6 bacterial genera (*Lactobacillus, unclassified_f__Oscillospiraceae, Ruminococcus_torques_group, norank_f__Peptococcaceae, Candidatus_Stoquefichus*, and *Family_XIII_AD3011_group*) enriched in the Fer group were positively correlated with BMD, blood calcium, OCN, Tb.Ar, and BV.TV, and negatively correlated with TRACP-5b, IL-6, TNF-α, and Tb.sp (Figure 10A). In contrast, the *g_Prevotellaceae_Ga6A1 group* showed the opposite correlation (Figure 10A).

A correlation analysis between the possible key metabolites and bone markers. The 17 serum metabolites enriched in the Fer group were positively correlated with BMD, blood calcium, OCN, Tb.Ar, and BV.TV, and negatively correlated with TRACP-5b, IL-6, TNF-α, and Tb.sp (Figure 10B). In contrast, the 23 serum metabolites downregulated in the Fer group showed the opposite correlation (Figure 10B).

A correlation analysis between possible key bacteria and possible key serum metabolites. The 6 bacterial genera enriched in the Fer group were positively correlated with the 17 serum metabolites enriched in the Fer group (Figure 10C). *g_Prevotellaceae_Ga6A1_group* was positively correlated with the 23 serum metabolites downregulated in the Fer group (Figure 10C).

## 4. Discussion

This study investigated the effects of honeysuckle fermented by LP on OP rats. Bone turnover is a continuous process coordinated by bone-synthesizing osteoblasts and bone-resorbing osteoclasts. Excessive osteoclastogenesis can break the dynamic balance between osteoclastic bone resorption and osteoblastic bone formation and lead to OP [32]. Rats were administered dexamethasone to establish an OP model. It fosters osteoclast formation and differentiation, while concurrently suppressing osteoblast activity [34]. This disparity accentuates bone resorption and curtails bone formation. Furthermore, it instigates various inflammatory responses and impedes calcium absorption in the intestines and kidneys [35,36]. These events collectively led to decreased bone density, a significant reduction in the size and number of trabeculae, severe destruction of the bone microarchitecture, decreased blood calcium levels, increased osteoclast-related differentiation products, such as TRACP-5b, and decreased osteoblast-related differentiation markers, such as OCN, in the circulatory system. However, our study showed that the OP characteristics were significantly improved in rats after the administration of an FS intervention.

Research on the gut–bone axis is a hot topic in OP research. The metabolites produced by GM (such as short-chain fatty acids, etc.), the nutrients absorbed (calcium, vitamin D, etc.), and the systemic immune response involved in the related lymphoid tissues all affect bone metabolism [17]. Many substances in FS cannot be directly absorbed by the small intestine, but can only be absorbed into the blood after fermentation by GM to exert the anti-OP effect. GM was closely related to OP, and restoring the balance of GM could improve OP [37]. In this study, we observed reduced bacterial richness and diversity in the colons of OP rats. Our study showed that 6 bacterial genera enriched in the colons of rats ingesting FS may contribute to the recovery of OP. Namely, *Lactobacillus*, *Oscillospiraceae*, *Ruminococcus_torques_group*, *Peptococcaceae*, *Candidatus stoquefichus*, and *Family_XIII_AD3011_group*, might have played significant roles in potentiating OP relief. *Lactobacillus* gained recognition for its therapeutic impact on OP. *Lactobacillus reuteri* demonstrated potential in mitigating bone loss in older women with low bone density [38]. *Lactobacillus acidophilus* inhibited bone loss and increased bone heterogeneity in OP mice by modulating the Treg-Th17 cell balance [39]. Genera like *Oscillospiraceae* and *Peptococcaceae*, known to produce Short-chain fatty acids (SCFA) SCFA in GM, not only served as an energy source for epithelial cells but also treated OP by inhibiting osteoclast differentiation [40]. Moreover, the *Oscillospiraceae* and *Candidatus stoquefichus* genera exhibited an inverse correlation with inflammation-triggering factors, such as TNF-α [41,42]. Menaquinone-7, used to treat bone health in rats, was enriched in the feces of the *Ruminococcus_torques_group* [43]. In addition, the presence of the *Family_XIII_AD3011_group* may have pointed toward a reduced risk regarding conditions such as Crohn’s disease and ulcerative colitis [44].

The 23 metabolites enriched in the serum of OP rats may be related to the pathogenesis of OP (Appendix A). SM(d18:1/16:1(9Z)) and SM(d18:2(4E,14Z)/16:0) are variants within the sphingomyelin family and play a pivotal role in normal bone turnover, as well as in preserving bone density [45]. The accumulation of sphingomyelin due to the hindered conversion process of sphingomyelin to ceramide has potential implications for bone health [46]. The levels of L-methionine and L-cysteine are positively correlated with homocysteine concentrations and, when elevated, may lead to decreased bone density [47,48]. Enterochromaffin cells, predominantly located in the gut, are the primary source of the synthesis of most of the body’s serotonin [49]. Once synthesized, this serotonin circulates peripherally and has the capacity to modulate bone tissues [50]. Clinical evidence suggests that elevated blood serotonin levels are associated with an inhibition in bone formation, leading to reduced bone mineral density [51]. In OP, the equilibrium in bone remodeling is disturbed, leading to disproportionate bone absorption in comparison with bone formation. Such a change may influence the regulation of serotonin synthesis, potentially elevating the levels of serotonin in the serum. The elevation in glutathione conjugates could potentially indicate the body’s strategy to counteract oxidative stress [52]. Such stress appears to deter osteoblast activity while simultaneously activating osteoclasts [53,54]. Simultaneously, this leads to an upswing in inflammation, contributes to weakening of the bone infrastructure, and ultimately results in overall bone loss [55]. We speculate that these 6 serum metabolic components (SM(d18:1/16:1(9Z)), SM(d18:2(4E,14Z)/16:0), L-methionine, L-cysteine, serotonin, and 4-Hydroxyestrone-2-S-glutathione) are the key metabolites causing bone metabolism disorders in rats in the Mod group.

There were 17 serum metabolites enriched in the Fer group (Appendix A). Palmatine improved cartilage degeneration and bone loss in OA–OP rats, and its serum metabolite 2-methoxyacetaminophen sulfate was also restored [56]. 2-Methoxyacetaminophen sulfate likely became one of the undiscovered markers of serum metabolites that improved OP. The contents of these 2 metabolites (2-methoxyacetaminophen sulfate and methylisocitric acid) in FS were significantly higher than those in MS. These 2 components are probably the active ingredients released after the fermentation of honeysuckle and are directly into the blood in the small intestine to exert anti-OP effects. Protocatechuic acid inhibited osteoclast differentiation and stimulated apoptosis in mature osteoclasts [57]. 3-hydroxybenzoic acid and its metabolic precursors were used to prevent and treat a variety of chronic diseases [58]. Daily low-dose oral administration of 3-hydroxybenzoic acid was also effective in improving stress resistance in laboratory rodents and had anti-inflammatory and analgesic activity in animal models of traditionally known NSAIDs. Indole-3-acetic acid had been reported to change the intestinal flora of mice and alleviate ankylosing spondylitis [59]. These 3 metabolites (protocatechuic acid, 3-hydroxybenzoic acid, and indole-3-acetic acid) appeared in both FS and MS. These 3 metabolites may be effective components in honeysuckle liquid or LP, are directly absorbed into the blood, and exert anti-OP effects. Ganglioside GT3 (d18:1/20:0) and Ganglioside GD2 (d18:1/12:0) are substances that are not contained in FS. Gangliosides played a role in bone formation [60]. Ganglioside GD2 served as a marker for the identification and purification of mouse bone marrow mesenchymal stem cells [61]. The sources of these 2 substances (Ganglioside GT3 (d18:1/20:0) and Ganglioside GD2 (d18:1/12:0)) may be substances in FS that are not absorbed by the small intestine but pass through the colon and enter the blood under the action of the gut–bone axis, entering the blood and exerting their effects. There are relatively few reports on the other 10 metabolites, and further research is needed on how they exert their anti-OP functions.

A correlation analysis was performed between the 6 bacterial genera and the 17 serum metabolites and the bone-related indices. The results showed that there was a positive correlation between these 6 bacterial genera and these 17 serum metabolites and positive bone markers. FS may alleviate OP by restoring the abundance of the 6 bacterial genera and the contents of the 17 serum metabolites to reduce osteoclast differentiation, promote osteoblast differentiation, and alleviate inflammatory response.

## 5. Conclusions

In conclusion, our results showed that FS may alleviate OP by restoring the levels of 6 bacterial genera and 17 serum metabolites to reduce osteoclast differentiation, promote osteoblast differentiation, and alleviate inflammatory response. The 6 genera are *Lactobacillus*, *unclassified_f__Oscillospiraceae*, *Ruminococcus_torques_group*, *norank_f__Peptococcaceae*, *Candidatus stoquefichus*, and *Family_XIII_AD3011_group*. The 17 serum metabolites are 2-methoxyacetaminophen sulfate, methylisocitric acid, protocatechuic acid, 3-hydroxybenzoic acid, 1-methyladenosine, indole-3-acetic acid, 2’,5’-dideoxyadenosine, DG(22:6(4Z,7Z,10Z,13E,15E,19Z)-OH(17)/i-12:0/0:0), 3,5-dichlorosalicylic acid, PC(20:3(8Z,11Z,14Z)-2OH(5,6)/2:0), ganglioside GT3 (d18:1/20:0), ganglioside GD2 (d18:1/12:0), perylene-1,2-dione, 12-hydroxydodecanoylcarnitine, 1-(2-amino-3-hydroxyphenyl)-ethanone sulfate, PE(20:4(8Z,11Z,14Z,17Z)/22:0), and D-threonine. Methylisocitric acid and 2-methoxyacetaminophen sulfate, likely the primary active ingredients in honeysuckle fermentation, directly enter the bloodstream in the small intestine to confer the anti-OP benefits. Similarly, protocatechuic acid, 3-hydroxybenzoic acid, and indole-3-acetic acid are the potential effective components in honeysuckle or LP that are absorbed into the blood for the same effect. Gangliosides GT3 (d18:1/20:0) and GD2 (d18:1/12:0) in FS bypass the small intestine but pass through the colon and enter the blood under the action of the gut–bone axis, entering the blood and exerting their effects.

## Figures and Tables

**Figure 1 foods-13-02649-f001:**
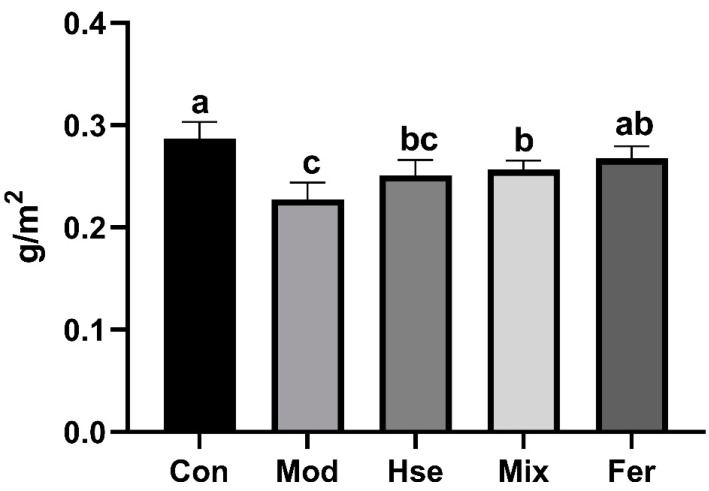
FS more significantly restored BMD in OP rats. The BMD of the femur was analyzed using dual-energy X-ray absorptiometry (Lunar iDXA, GE). Values are given as mean ± SD (*n* = 6). Values with different superscript letters are significantly different at *p* < 0.05.

**Figure 2 foods-13-02649-f002:**
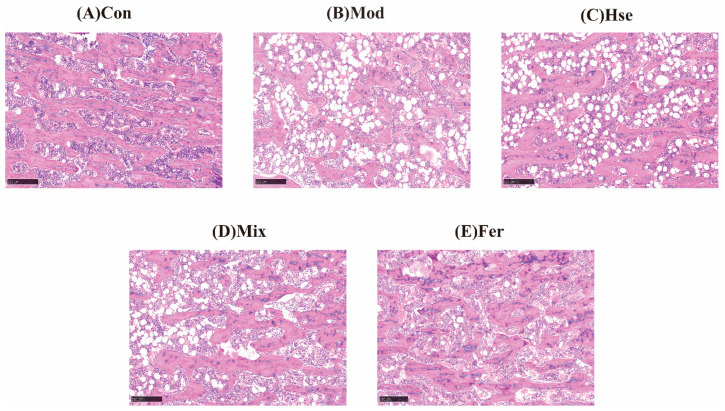
FS repaired bone microstructural damage in osteoporotic rats. Representative pathological sections of the Con group (**A**), Mod group (**B**), Hse group (**C**), Mix group (**D**), and Fer group (**E**). The Con, Mod, Hse, Fer, and Mix groups’ right tibias were sampled and prepared for paraffin sectioning, H&E staining, and bone microarchitecture. The photo magnification is 10 times.

**Figure 3 foods-13-02649-f003:**
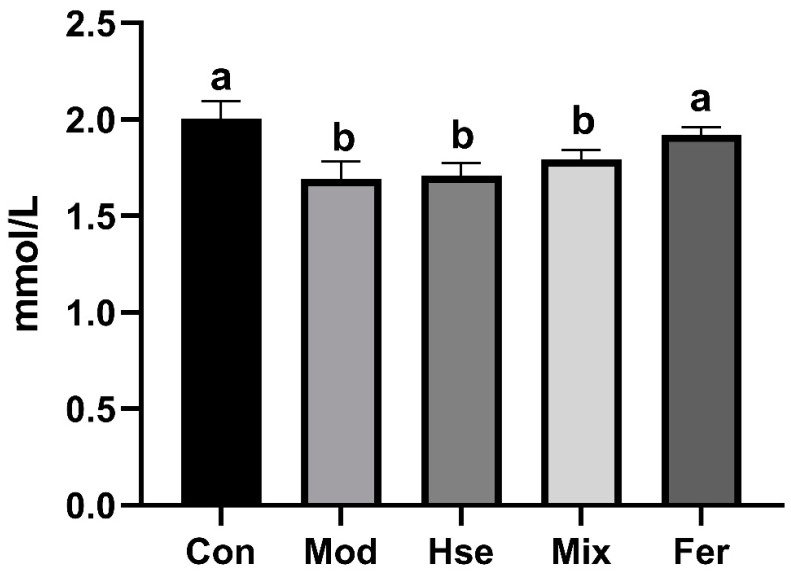
FS more significantly restored blood calcium levels in osteoporotic rats. Values are expressed as mean ± SD (*n* = 6). Values with different superscript letters are significantly different at *p* < 0.05.

**Figure 4 foods-13-02649-f004:**
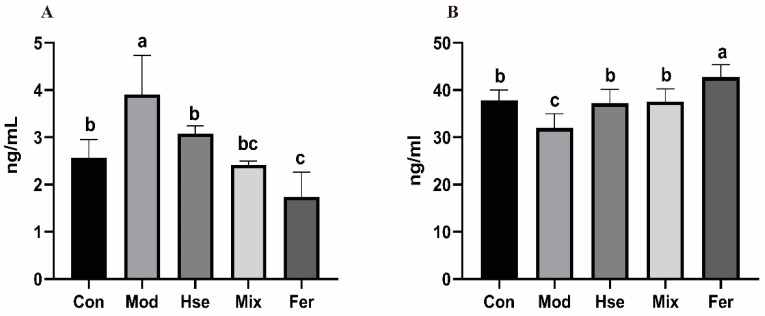
Serum levels of TRACP-5b (**A**) and OCN (**B**). The levels of TRACP-5b and OCN in rat serum were measured using ELISA kits. Values are expressed as mean ± SD (*n* = 6). Values with different superscript letters are significantly different at *p* < 0.05.

**Figure 5 foods-13-02649-f005:**
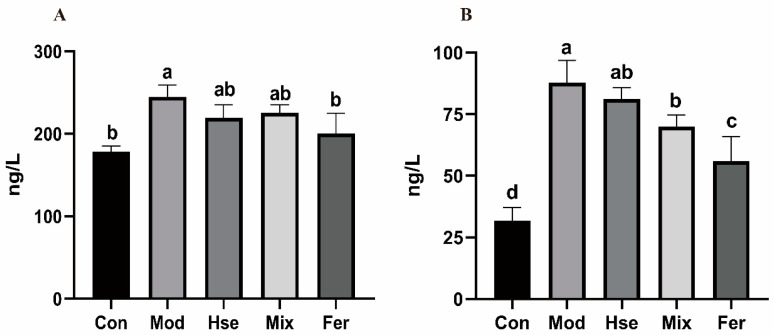
Serum levels of TNF-α (**A**) and IL-6 (**B**). The TNF-α and IL-6 levels in rat serum were measured using ELISA kits. Values are expressed as mean ± SD (*n* = 6). Values with different superscript letters are significantly different at *p* < 0.05.

**Figure 6 foods-13-02649-f006:**
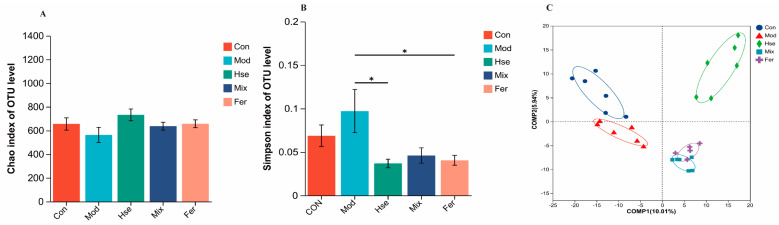
Microbial diversity analysis of colon contents of rats in 5 groups: Chao 1 index (**A**), Simpson index (**B**), PLS-DA analysis (**C**). Data are means ± SD and analyzed by one-way ANOVA. * *p* < 0.05 (*n* = 6).

**Figure 7 foods-13-02649-f007:**
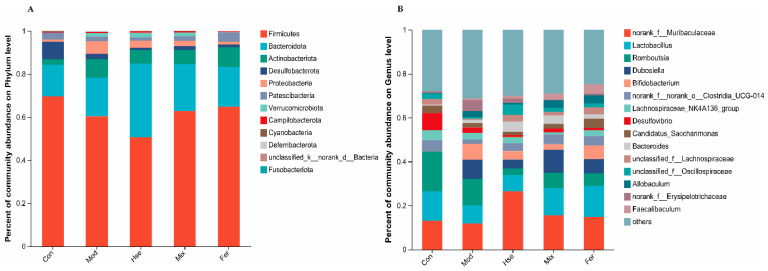
Composition of the GM at the phylum (**A**) and genus (**B**) levels. The relative abundance of different groups of GM was analyzed at the genus level to identify potential key bacteria. *p* < 0.05 (*n* = 6).

**Figure 8 foods-13-02649-f008:**
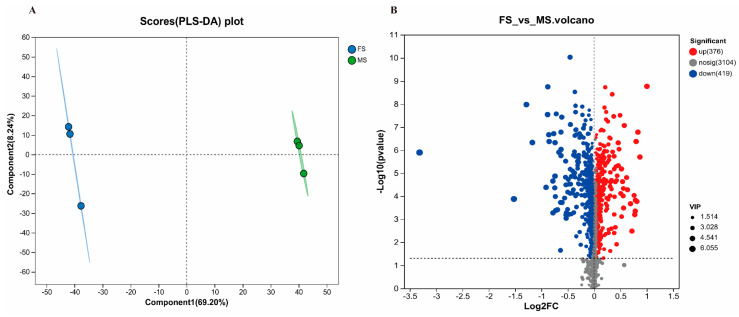
Metabolic composition and blood composition analysis of FS and MS. PLS-DA analysis of FS and MS (**A**); difference volcano plot analysis of FS and MS (**B**); *p* < 0.05 (*n* = 6).

**Figure 9 foods-13-02649-f009:**
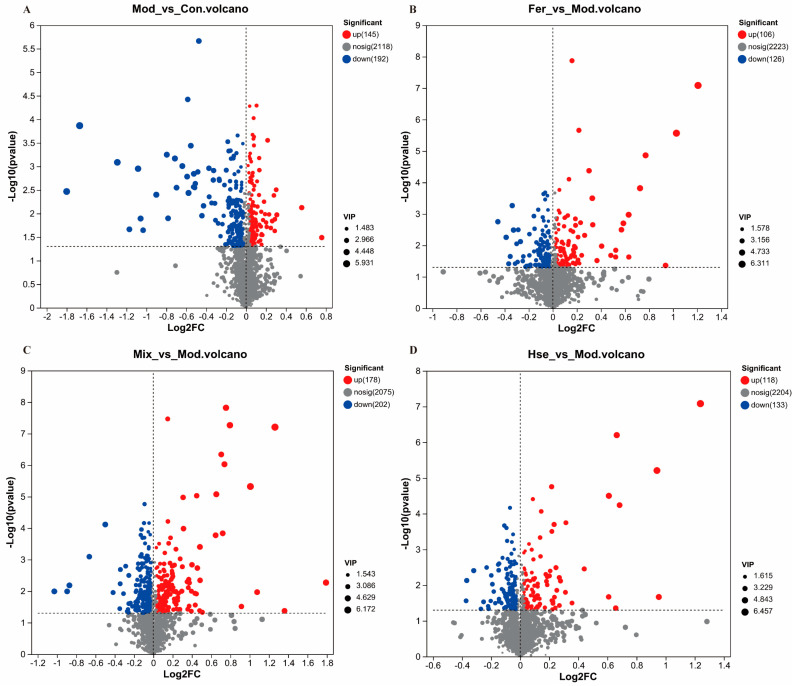
Analysis of differential metabolites in serum components. Analysis of differential metabolites in the Con group and Mod group (**A**); analysis of differential metabolites in the Fer group and Mod group (**B**); analysis of differential metabolites in the Mix group and Mod group (**C**); analysis of differential metabolites in the Hse group and Mod group (**D**). *p* < 0.05 (*n* = 6).

**Figure 10 foods-13-02649-f010:**
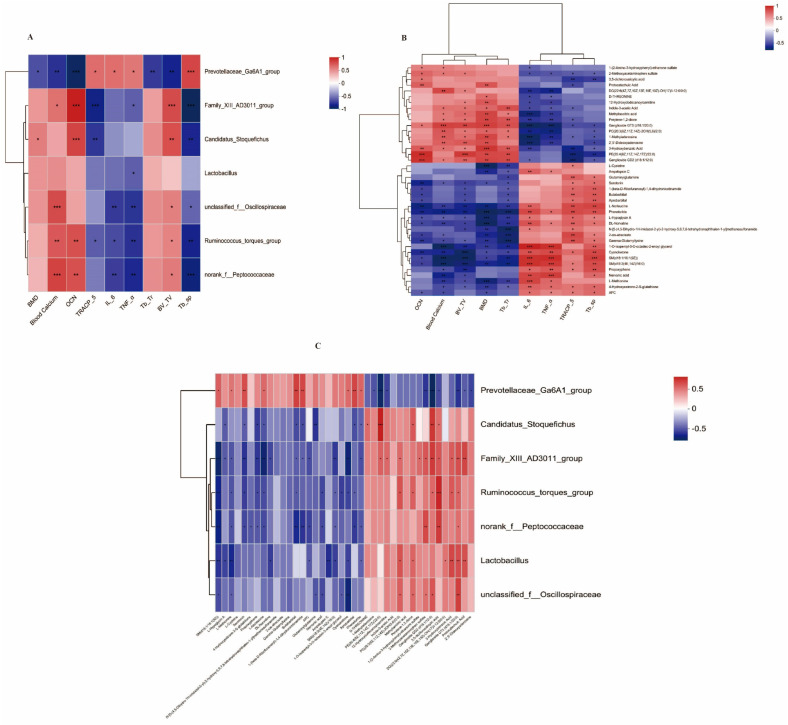
Correlation analysis. Correlation analysis between potential key bacteria and OP-related indicators (**A**), correlation analysis of key metabolites and OP-related indicators (**B**). Correlation analysis between potential key bacteria and potential key metabolites (**C**). * *p* < 0.05; ** *p* < 0.01; *** *p* < 0.001 (*n* = 6).

## Data Availability

The original contributions presented in the study are included in the article/Appendix A, further inquiries can be directed to the corresponding author.

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
