# Peer review of "Lonicera japonica Fermented by Lactobacillus plantarum Improve Multiple Patterns Driven Osteoporosis"

_foods, 2024, doi:10.3390/foods13172649_

Round 1

Reviewer 1 Report

Comments and Suggestions for Authors

Dear Authors,

I have read and reviewed the manuscript foods-3122402, Multiple Patterns Driven Osteoporosis Improvement of Lonic- 2 era japonica Fermented by Lactobacillus plantarum, which discusses the role of gut microbiota in the positive effects of honeysuckle on osteoporosis in an animal model. The study is well planned and conducted, the paper is clearly conceived and written. My suggestion is to publish it after minor corrections.

Abstract

Line 9. Put a full stop after the word challenge. Start new sentence with Certain functional food has potential to…

Line 11.  Add Lonicera japonica between the words honeysuckle and solution.

1.       Introduction

Line 46. Write (GM) after word microbiota

Line 48. At the beginning of a sentence, write whole words gut microbiota, instead of an abbreviation GM.

Line 53. Do not use abbraviations (GM) at the beginning of a sentence.

Line 54. Does GM, itself, produce estrogen?

2.       Materials and methods

Line 86, The abbreviation MS should first be fully defined.

Line 96. Does administrated mean by orally gavage? It is not clear.

3.       Results

Line 242. The magnification at which the photos  were made should be specified in the figure legend.

4.       Discussion

Line 409-410.  …that these 6bacterial genera may be FS entering the gut… Unclear meaning. Please write more clearly.

Line 417. I see no connection between the mention of fluoxetine and the here presented results. Maybe that reference should be omitted?

Line 460. Start a sentence with Among these 17 serum…

Author Response

We are very grateful to the reviewer for your comments and revisions.

Our response is as follows:

Abstract:

“Line 9. Put a full stop after the word challenge. Start new sentence with Certain functional food has potential to…”has been revised to “Osteoporosis (OP) represents a global health challenge. Certain functional food has potential to mitigate OP.“

"Line 11.  Add Lonicera japonica between the words honeysuckle and solution. "  has been revised to "Honeysuckle (Lonicera japonica) solution"

introduction

"Line 46. Write (GM) after word microbiota"   has been revised to  "Gut microbiota(GM)"

"Line 48. At the beginning of a sentence, write whole words gut microbiota, instead of an abbreviation GM."   has been revised to   "Gut microbiota(GM), a mix of various microorganisms in the human gut, is vital for health and disease[16]"

"Line 53. Do not use abbraviations (GM) at the beginning of a sentence. ”    has been revised to   “Gut microbiota also engages in endocrine interactions, producing hormonal substances like 5-HT and estrogen, which are essential for bone metabolism regulation[22,23].”

“Line 54. Does GM, itself, produce estrogen?”   reply:“GM cannot directly produce estrogen”  has been revised to “Gut microbiota also engages in endocrine interactions, producing hormonal substances like 5-HT and estrogen”

2. Materials and methods

“Line 86, The abbreviation MS should first be fully defined.”  has been revised to  “ In order to analyze the differences between honeysuckle before and after fermentation. An unfermented solution, namely mixed solution (MS), was used as a control.

“Line 96. Does administrated mean by orally gavage? It is not clear.”  reply:“Administration was by gavage”  has been revised to “The Hse group was orally gavaged with the honeysuckle extract solution at a dosage of 1 mL/100 g daily. Similarly, the Fer group was orally gavaged with the fermentation solution at the same dosage of 1 mL/100 g daily. The Mix group was orally gavaged with a mixed solution ”

3.Results

“Line 242. The magnification at which the photos  were made should be specified in the figure legend.”  has been revised to “The Con, Mod, Hse, Fer, and Mix groups’ right tibias were sampled and prepared for paraffin sectioning, H&E staining, and bone microarchitecture. The photo magnification is 10 times.“

Discussion

”Line 409-410.  …that these 6bacterial genera may be FS entering the gut… Unclear meaning. Please write more clearly.“  reply:We deleted this sentence

”Line 417. I see no connection between the mention of fluoxetine and the here presented results. Maybe that reference should be omitted?“  has been revised to    ”Accumulation of sphingomyelin due to the hindered conversion process of sphingomyelin to ceramide, with potential implications for bone health[46]. “

”Line 460. Start a sentence with Among these 17 serum…“  has been revised to ”The 17 serum metabolites enriched in the Fer group “

The modifications I made to the article are indicated in red within the attachment.
Appreciate the valuable comments providedby the reviewer.

Reviewer 2 Report

Comments and Suggestions for Authors

All suggestions and questions are provided throughout the manuscript.

Comments on the Quality of English Language

All suggestions and questions are provided throughout the manuscript.

Author Response

The modifications l made to the article areindicated in red within the attachment.
Appreciate the valuable comments providedby the reviewer.

Round 2

Reviewer 2 Report

Comments and Suggestions for Authors

The authors satisfied my suggestions and questions in the manuscript.
